# A novel theoretical framework for simultaneous measurement of excitatory and inhibitory conductances

**Daniel Müller-Komorowska**[1,2], **Ana Parabucki**[3], **Gal Elyasaf**[3], **Yonatan Katz**[3], **Heinz Beck**[1], **Ilan Lampl**[3]*

**1** Institute of Experimental Epileptology and Cognition Research, Life and Brain Center, University of Bonn Medical Center, Bonn, Germany, **2** International Max Planck Research School for Brain and Behavior, University of Bonn, Bonn, Germany, **3** Department of Neurobiology, Weizmann Institute of Science, Rehovot, Israel

* ilan.lampl@weizmann.ac.il

**Data Availability Statement:** All code and most of the simulation results are available at https://

## Abstract

The firing of neurons throughout the brain is determined by the precise relations between excitatory and inhibitory inputs, and disruption of their balance underlies many psychiatric diseases. Whether or not these inputs covary over time or between repeated stimuli remains unclear due to the lack of experimental methods for measuring both inputs simultaneously. We developed a new analytical framework for instantaneous and simultaneous measurements of both the excitatory and inhibitory neuronal inputs during a single trial under current clamp recording. This can be achieved by injecting a current composed of two high frequency sinusoidal components followed by analytical extraction of the conductances. We demonstrate the ability of this method to measure both inputs in a single trial under realistic recording constraints and from morphologically realistic CA1 pyramidal model cells. Future experimental implementation of our new method will facilitate the understanding of fundamental questions about the health and disease of the nervous system.

## Author summary

Most neurons in the brain receive synaptic inputs from both excitatory and inhibitory neurons. Together, these inputs determine neuronal activity: excitatory synapses shift the electrical potential across the membrane towards the threshold for generation of action potentials, whereas inhibitory synapses lower this potential away from the threshold. Action potentials are the rapid electrochemical signals that transmit information to other neurons and they are critical for the information processing abilities of the brain.

Although there are many ways to measure either excitatory or inhibitory inputs, these methods have been unable to measure both at the same time. Measuring both inputs together is essential towards understanding how neurons integrate information. We developed a new analytical method to measure excitatory and inhibitory inputs at the same time from the voltage response to injection of an alternating current into a neuron. We describe the foundation of this new method and find that it works in biologically

github.com/danielmk/ENCol. The data not included in the repository can be simulated with the provided code.

**Funding:** This work was supported by grants from the DFG-SFB 1089 (to HB and IL), 01EW1606 - DeCipher EraNet Neuron (to HB and IL), Israel Science Foundation (ISF 1539/17 and ISF Bikura 2799/20) (to IL), Minerva Foundation (to IL) and Human Frontier Science Program (HFSP) (to IL). IL is the incumbent of the Norman and Helen Asher Professorial Chair. The funders had no role in study design, data collection and analysis, decision to publish, or preparation of the manuscript.

**Competing interests:** The authors have declared that no competing interests exist.

realistic simulations of neurons. By using this technique in real neurons, scientists could investigate basic principles of information processing in the healthy and diseased brain.

## Introduction

Neuronal firing is orchestrated by the interplay of excitatory and inhibitory inputs. Therefore, studying their relationship has been crucial to solving fundamental questions in cellular and system neuroscience. Disrupted relations between these inputs were suggested to accompany many neurological diseases and in particular epileptic seizures. It is commonly believed that such seizures are accompanied and even caused by a disruption of excitation-inhibition ratio and their temporal relationships [1–3].

The most widely used method to measure inhibitory and excitatory inputs in isolation is the voltage clamp technique. To reveal excitatory synaptic currents the membrane potential is voltage clamped near the reversal potential of inhibition (near -80 mV) and inhibitory synaptic currents are revealed when the voltage is clamped near the excitatory reversal potential (near 0 mV). Voltage clamp recordings have been used in this manner to study mechanisms of feature selectivity of cortical cells belonging to various modalities [4–13]. Current clamp recordings also allow for the isolation of excitatory and inhibitory conductances, which is done by injecting constant positive or negative currents which bring the membrane potential near the reversal potential of these two input types [8–10,14–18].

Voltage and current clamp approaches share several similarities. In both cases, excitation and inhibition are recorded in different trials and conductances are estimated by fitting the averaged data with the membrane potential equation (Eq 1 below). Hence, these methods provide only an average picture and thus fail to capture the instantaneous and trial-by-trial based insight into the relations between excitation and inhibition.

The instantaneous relation between excitation and inhibition in-vivo was revealed using a different approach, relying on the finding that the membrane potential of nearby cortical cells in anesthetized animals is highly synchronized [19,20]. This approach consists of depolarizing one cell to reveal its inhibitory inputs while simultaneously hyperpolarizing a neighboring cell to reveal its excitatory inputs. Doing this showed that excitatory and inhibitory synaptic inputs are highly correlated in anesthetized and awake rodents [21,22] and was used to study the degree of correlation during oscillatory neuronal activities [23]. However, this approach depends on making the recordings from highly correlated cells, mostly observed in deeply anesthetized animals. Methods for estimation of excitatory and inhibitory inputs of a single cell during single trials were previously developed [24–28]. However, these methods make significant assumptions about the dynamics and statistics of the inputs. Importantly, all these methods rely on the occurrence of membrane potential fluctuations when estimating excitatory and inhibitory conductances. Clearly, changes in conductance sometimes are not accompanied by any change in membrane potential, as expected when a cell receives tonic shunting synaptic input with a reversal potential near the resting potential of the cell.

We describe a new theoretical framework for simultaneously measuring both excitatory and inhibitory conductances under current clamp in a single trial with high temporal resolution, without making statistical assumptions about the inputs. It is based on frequency analysis of the response of neurons when injected with a current composed of two sinusoidal components and allows measuring both the excitatory and inhibitory conductances simultaneously with membrane potential as a function of time. We demonstrate this method in-silico using simulations of a point neuron receiving excitatory and inhibitory synaptic inputs as well as in

a realistic pyramidal cell model when synapses are distributed further away from the soma. Finally, we describe the limitations of this approach in whole cell patch clamp recordings obtained using contemporary intracellular amplifiers.

## Results

### Transformation of membrane potential and total conductance to E and I conductances

We sought to develop a method that provides a way to simultaneously measure the excitatory and inhibitory conductances in a single trial with high temporal resolution during current clamp recording. We begin with the membrane Eq 1 for passive synaptic inputs of a point neuron, which can be rearranged to isolate the excitatory and inhibitory conductances as shown in Eq 2.

$$C \cdot \frac{dV(t)}{dt} = -(g_l(V(t) - V_l) + g_e(t)(V(t) - V_e) + g_i(t)(V(t) - V_i) - I(t)) \qquad (1)$$

Replacing $V(t) - V_l$, $V(t) - V_e$, $V(t) - V_i$ with $V^l(t)$, $V^e(t)$, $V^i(t)$ respectively and assuming that the total conductance equals the sum of the inhibitory and excitatory conductance $g_s(t) = g_i(t) + g_e(t)$ we get:

$$g_e(t) = \frac{C \cdot \frac{dV(t)}{dt} + g_l \cdot V^l(t) + g_s(t) \cdot V^i(t) - I(t)}{(V^i(t) - V^e(t))}; \qquad g_i(t) = g_s(t) - g_e(t) \qquad (2)$$

Eq (2) shows that the two inputs can be isolated if the following parameters are known: $V(t)$, membrane voltage; $g_l$, leak conductance; $g_s(t)$, total synaptic conductance; $V_l$, $V_e$, $V_i$, equilibrium potentials of the individual conductances; $C$, membrane capacitance; $I$, stimulus current. Fig 1 shows how this equation works in a simulated point neuron where these parameters

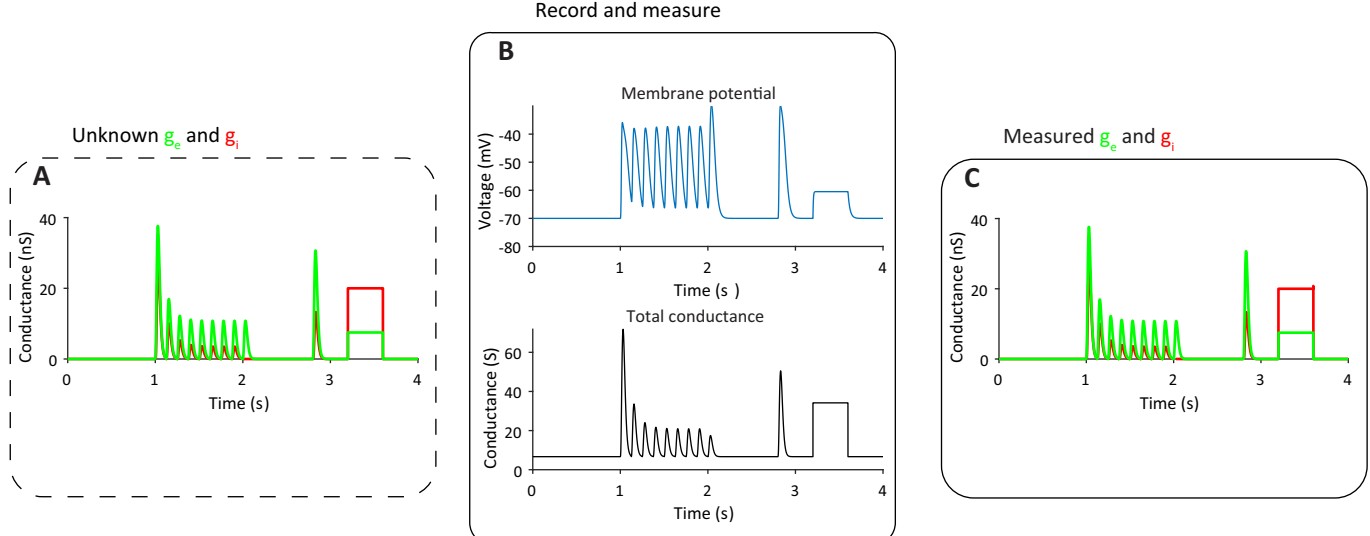

**Fig 1. G$_e$ and g$_i$ can be obtained from V(t) and total g. A.** Simulation of correlated excitatory (green) and inhibitory (red) synaptic inputs (inhibition delayed by 4 ms after excitation), which depressed according to a mathematical description of short-term synaptic depression (STD, Markram and Tsodyks, 1997). These are the inputs the method aims to reveal. **B.** Membrane potential simulation of a passive point neuron (R = 300MΩ, C = 0.15nF, Euler method, dt = 0.0005s) receiving the inputs in A, with the total conductance shown below. We assume that these two vectors are measurable. A short test current pulse was injected at the early part of the trace. **C** The result of transforming V(t), its derivative (not shown) and the total synaptic conductance into g$_e$ and g$_i$ using Eqs (1 and 2).

are indeed known. We demonstrate this transformation by showing depressing excitatory and inhibitory inputs as well as a step change in conductance. However, it works for any type and dynamic of excitatory and inhibitory inputs.

How do we find these parameters under experimental conditions? The equilibrium potentials are generally assumed to be known and determined from intracellular and extracellular ion concentrations. The leak conductance and membrane capacitance can be measured when injecting hyperpolarizing current steps. The voltage is also easy to resolve during the current clamp. However, developing a method to record the membrane potential and at the same time also measure the conductance at each time point has been challenging. As we describe below, we can theoretically estimate the total conductance of the cell by measuring the voltage response during injection of a current composed of two high-frequency sinusoidal components. We start with impedance analysis of passive circuits representing a simplified point neuron with a patch clamp pipette and describing the relationships between the impedance and cell conductance.

## Impedance-conductance relationship in a passive point neuron

To develop a method that can be practically used for whole cell patch recordings, we included the resistance of the patch pipette in our analysis. As shown below, the resistance of the electrode affects the measurement of the cell's impedance and thus cannot be ignored. We analyzed in the frequency domain the impedance of a circuit composed of a recording electrode ($R_s$) and a simplified point neuron (composed of a conductance, $g(t)$ (equal to $g_l+g_e(t)+g_i(t)$) and a capacitor, $C$). The impedance of this circuit is given by Eq 3 ($w = 2\pi f$, $j$ is the imaginary unit and $f$ is the frequency in Hertz). The cell conductance ($g(t)$) and the pipette resistance ($R_s(t)$ can vary over time, and so consequently also the impedance of the circuit ($Z(t)$).

$$Z(f,t) = R_s(t) + \frac{1}{(g(t) + j \cdot w \cdot C)} = R_s(t) + \frac{g(t)}{g(t)^2 + (w \cdot C)^2} - \frac{j \cdot w \cdot C}{g(t)^2 + (w \cdot C)^2} \qquad (3)$$

Fig 2 illustrates the relationships between the impedance and $g$ for various frequencies (for constant values). It also shows that in the presence of $R_s$, impedance-frequency curves intersect each other as frequency increases, resulting in a positive relationship between circuit impedance and $g$ for a large range of $g$ (compare Fig 2A and 2C). The presence of $R_s$ also keeps the phase almost constant for different frequencies and $g$ values Fig 2D). Thus the electrode resistance has a prominent effect on the total impedance of this circuit and should not be ignored when injecting high frequency sinusoidal current into cells.

## The in-silico experiment

In the next sections, we show the response of a point neuron to an injection of a current (Fig 3D) composed of two sinusoidal components (Eq (4), $w_1 = 2\pi f_1$, $w_2 = 2\pi f_2$):

$$I(t) = I_1 \cdot sin(w_1 \cdot t) + I_2 \cdot sin(w_2 \cdot t) \qquad (4)$$

can be used to measure changes in excitatory and inhibitory conductances imposed on the model (Fig 2B) in a single trial. Although the voltage response in our simulation fluctuates across a large range of more than 35mV (Fig 3C), most of the drop of voltage occurs on the electrode resistor, as seen when we set $R_s$ to zero (Fig 3E). Due to the low-pass filtering of the input by the passive properties of the cell when injecting high frequency sinusoidal current, the fluctuations of the voltage across the membrane itself are extremely attenuated, resulting in less than 6mV peak to peak amplitudes. Such small fluctuations are unlikely to recruit any voltage-gated intrinsic current. Note that the value of the electrode resistance accounts for both

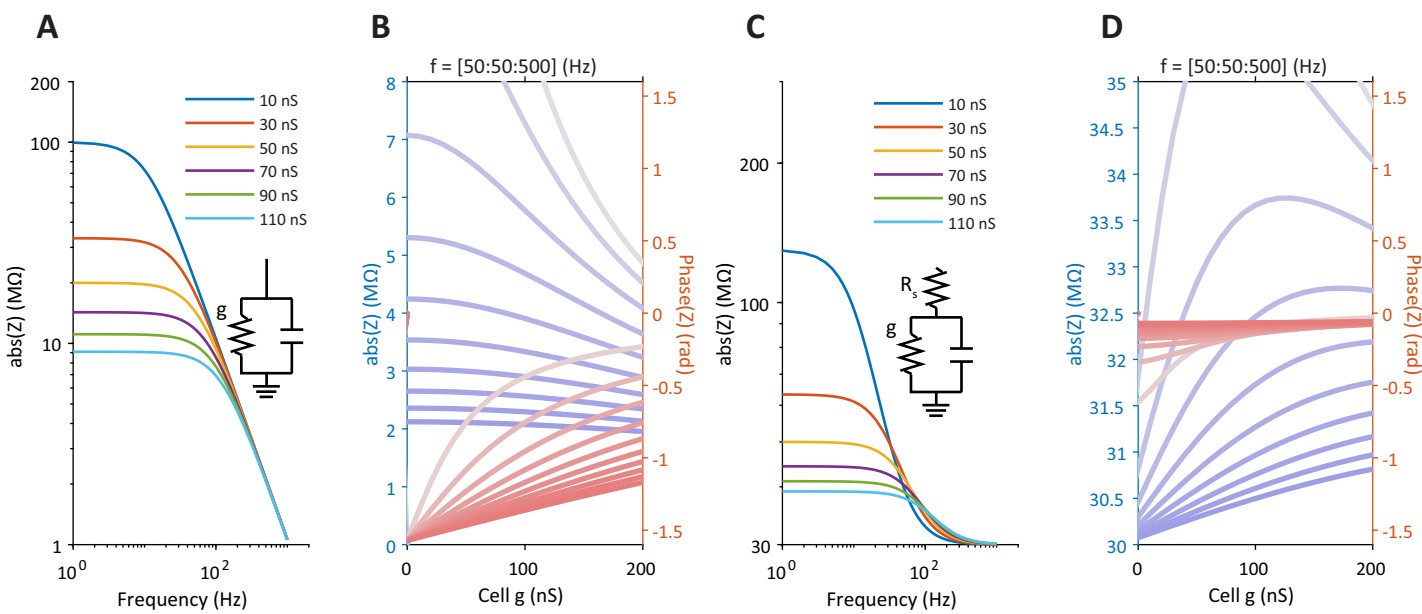

**Fig 2. Impedance frequency-curves of passive electrical circuits for different conductances. A.** Absolute impedance as a function of frequency for different values of the model conductance. Note that none of the curves intersect. **B.** Absolute impedance curves as function of conductance together with phase curves between real and imaginary parts of the impedance. Each line represents a different frequency (50Hz to 500Hz, steps of 50Hz, from lowest (pale blue or red) to highest (deep colors) as indicated by the text (Fr = [50:50:500]) above. Also presented are phase curves between voltage and current for the same frequencies. **C-D.** The same but when the RC circuit is also connected in series to a resistor ($R_s$). Note in c that curves intersect each other at high frequencies and in d that the phase is almost constant. Fixed circuit parameters: $R_s$ = 30MΩ, C = 0.15nF.

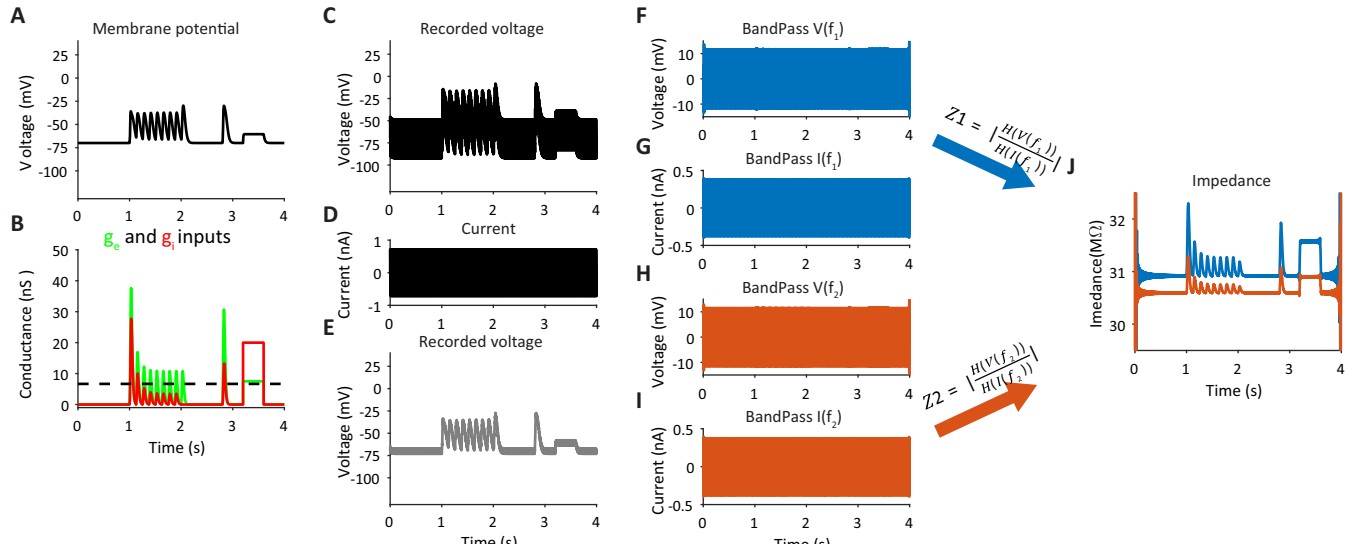

**Fig 3. Measurement of total impedance in a single trial -simulation of a point neuron. A.** Simulated membrane potential of a point neuron when receiving synaptic conductances as shown in **B.** (excitation–green, inhibition–red). **C-D.** The voltage response of a simulated neuron I receiving synaptic inputs described in b and injected with a current composed of two sinusoidal components (d, 0.375nA, 210Hz and 315Hz). 'Recording' was made via an electrode of 30MΩ and thus most of the voltage drop due to the injected current occurred across the electrode. **E.** The actual voltage change across the membrane was small (as 'recorded' when electrode resistance was set to zero). **F-I.** voltage and current traces when filtered at the two frequencies used to compose the current. Note for the small fluctuations in voltage. **J** Impedance curves for each of the two frequencies obtained by dividing the Hilbert transform of the voltage and current shown in F-I and then taking the absolute values. Edge effect of the filtering is observed near zero and end times of the traces.

the pipette and access resistance. In our our simulation we set the electrode resistance to 30MΩ, which is higher than for the typical access resistance in in-vitro recordings, but well within the range of in-vivo recordings [29]. The current and the voltage are used to calculate all the passive properties of the simulated cell in a single trial (i.e., $R_s(t)$, $g$(t) and $C$). The computations are all analytical and approximation is done only when estimating the cell's capacitance as shown below. As described above, estimating the cell's conductance allows us to measure the excitatory and inhibitory conductances.

## Measurement of the cell's total conductance

The first step towards measuring the cell's excitatory and inhibitory conductances using injection of sinusoidal currents is to measure its total capacitance. The cell's capacitance is usually estimated from the response to a step current. Other methods for such estimation are also available, such as using a short pulse [30,31] and variance analysis of the response to injection of noise [32]. Here we show that a cell"s capacitance can be well estimated from the response to either one of the two frequencies composing the sinusoidal current (Eq (4)). We rely on the assumption that when the frequency of the current is high ($w^* C >> g(t)^2$), we can neglect $g(t)^2$ in the denominators of the second and third terms in Eq 3. Hence, at such frequencies the electrode resistance ($R_s$) is relatively larger than the second term, and thus the second term can be neglected. In this case, the total impedance of the circuit is mostly determined by the electrode resistance and the capacitance of the cell, as the latter draws most of the sinusoidal current that is injected into the cell. Here we ignore any stray capacitance in the recording system, such as of the recording pipette, but below we show that this capacitance can be partially compensated offline. The capacitance of the cell can be estimated from the voltage amplitude and phase relationship between the voltage and the current. These relationships can be approximated by Eq (5) (see also the phase curves in Fig 2D) obtained from Eq 3 when $w^*C >> g$.

$$Z(f) \approx R_s - \frac{j}{w \cdot C} \qquad (5)$$

For such an estimation to be valid (i.e., deriving Eq (5) from (3)), the frequency of each one of the two current components has to be sufficiently high. For example, for a cell with a mean conductance of 1/100MΩ and total capacitance of 0.15nF, recorded with 10MΩ electrode ($R_s$), a ratio of ~88 between $(w^*C)^2$ and $g^2$ will be obtained at 100Hz. Since the impedance of the second term in Eq (3) for this example is ~1MΩ, much smaller than $R_s$ (10MΩ), we neglect this term. Thus, the capacitance can be obtained from Eq (5), if we can estimate the electrode resistance and the phase relationship between the current and the voltage. We do this in a single trial when sinusoidal current is injected, by first measuring the electrode resistance ($R_{s,est}$) from the ratio of the absolute values of the fast Fourier Transform (FFT) of the voltage and the current at the frequency of the injected current, after both traces were bandpass filtered at one of the two frequencies (F1 or F2, using 'bandpass' Matlab function, implementing finite impulse response (**FIR**) filter). Importantly, this calculation is performed for a time window within which no stimulation is delivered (e.g., 1 second before stimulation). The two vectors (*FV*, *FI* bandpass filtered voltage and current) are then used to estimate $R_s$. For the measurement of the capacitance we provide a rough estimation of $R_s$, denoted with an asterisk. A more precise estimation of $R_s$ is provided later.

$$R_{s,est}^* = abs(fft(FV)/abs(fft(FI)); \qquad \text{(at F1 or F2)} \qquad (6)$$

The phase between FV and FI is calculated from the Hilbert transform of FV (H operator,

either for the F1 or F2) using the 'hilbert' Matlab function and averaging over time:

$$\theta_{est} = \overline{angle(H(FV)) - angle(H(FI))} \tag{7}$$

Averaging is performed for the same time window as above, within which no stimulation is delivered (e.g., 1 second before). The trigonometric relationships between the real and imaginary parts in Eq (5) are described in Eq (8), allowing to estimate the cell's total capacitance given that $R_s$ and $\theta_{est}$ are measured as described in Eqs (6) and (7):

$$C_{est} = 1/abs(tan(\theta_{est}) \cdot R^*_{s,est} \cdot w) \tag{8}$$

In the example shown in Figs 2 and 3, the real capacitance was set to 0.15nF and was estimated as 0.149nF. Note, that estimation of $C$ can also be obtained when setting $R_s$ to zero at a similar accuracy.

We then use the estimated capacitance of the cell to measure the cell's conductance and to obtain a more accurate measurement of the electrode resistance, both over time in a single trial. In this computation these values will be measured based on the analytical solution of Eq 3, this time without making any approximations. Here we use the fact that the current contains two sinusoidal components having two different frequencies (F1 and F2, e.g., 210Hz and 315Hz as used in the example). Since $Z(f)$ decreases with increasing frequency (Fig 2), increasing the frequencies, although it allows higher temporal resolution, will reduce the signal to noise ratio in the presence of noise. The voltage and the current are then bandpass filtered at the two frequencies (Fig 3F–3I, due to screen resolution are displayed as patches of colors). Note the small modulations in the bandpass filtered voltage signals, which are in the order of about 1mV. These modulations result from changes in the cell's conductance during the simulation of the synaptic inputs following the relationships between them as shown in Fig 2. For each bandpass filtered voltage and current trace: $FV_1(t)$, $FV_2(t)$, $FI_1(t)$, $FI_2(t)$ we computed the-hilbert transforms ($HFV_1(t)$, $HFV_2(t)$, $HFI_1(t)$, $HFI_2(t)$, using the 'hilbert' Matlab function). These complex vectors are then used to calculate the impedance of the cell at the two frequencies over time:

$$Z_1(f_1, t) = HFV_1(t)/HFI_1(t) \tag{9}$$

$$Z_2(f_2, t) = HFV_2(t)/HFI_2(t) \tag{10}$$

The absolute values of these complex vectors, shown in Fig 3J, demonstrate curves with a shape that is similar to that of the total conductance of the cell (leak plus synaptic conductances). Note that when the conductance of the cell is increased during activation of these inputs, the impedance is also elevated. This only happens in the presence of $R_s$, as shown in Fig 2.

These two impedance vectors are then used together to solve Eq 3 and obtaining a solution for $R_s(t)$ and $g(t)$ (when $z_1 \neq z_2$, $C$ is the estimated capacitance). To this end we used Mathematica (Wolfram) to solve the two equations for absolute values of $z_1$ and $z_2$ ("**Solve[Abs (r + 1/(g + I\*w1\*c)) = = Abs (z1) && Abs (r + 1/(g + I\*w2\*c)) = = Abs (z2), {r, g}]**", I = imaginary unit in Mathematica (Wolfram)) which gives the following solutions for $R_s$ and $g$ (here $Z_1$ and $Z_2$ are complex time dependent vectors, $j$ is the imaginary unit, and $C$ is the estimated

capacitance):

$$R_{s,est}(t) = (1/(2 \cdot j \cdot c(w_1 - w_2))) \cdot (j \cdot c(w_1 \cdot Z_1 - w_2 \cdot Z_1 + w_1 \cdot Z_2 + w_2 \cdot Z_2) + ((-j \cdot c(w_1$$
$$\cdot Z_1 + w_2 \cdot Z_1 - w_1 \cdot Z_2 + w_2 \cdot Z_2)^2 - 4 \cdot j \cdot c(w_1 - w_2)(Z_1 - Z_2 + j \cdot c \cdot w_1 \cdot Z_1 \cdot Z_2$$
$$- j \cdot c \cdot w_2 \cdot Z_1 \cdot Z_2))^{0.5}) \tag{11}$$

$$g_{est}(t) = -j \cdot (j + c \cdot R_{s,est}(t) \cdot w_1 - c \cdot w_1 \cdot Z_1)/(R_{s,est}(t) - Z_1) \tag{12}$$

In Eqs (11) and (12) $z_1$, $z_2$ as well as $R_{s,est}(t)$ are time dependent variables. Identical estimation will be obtained in Eq 12 after replacing $w_1$ and $z_1$ with $w_2$ and $z_2$. In Fig 4A, we again plotted the two impedance curves and also included the electrode resistance ($R_{s,est}(t)$), which is only slightly larger than its real value used in the simulation. The estimated total conductance is plotted in Fig 4C. Note that the estimated total conductance is almost identical in shape and magnitude to the sum of the leak, excitatory and inhibitory conductances used to simulate the membrane potential in this example.

## Estimation of the excitatory and inhibitory conductances from cell's conductance and membrane potential

After estimating the total conductance, Eqs (1) and (2) are used to compute the excitatory and inhibitory conductances as discussed above. Since sinusoidal current is injected into the cell (with two frequency components) we bandstop filter around each frequency (+- 5Hz) to

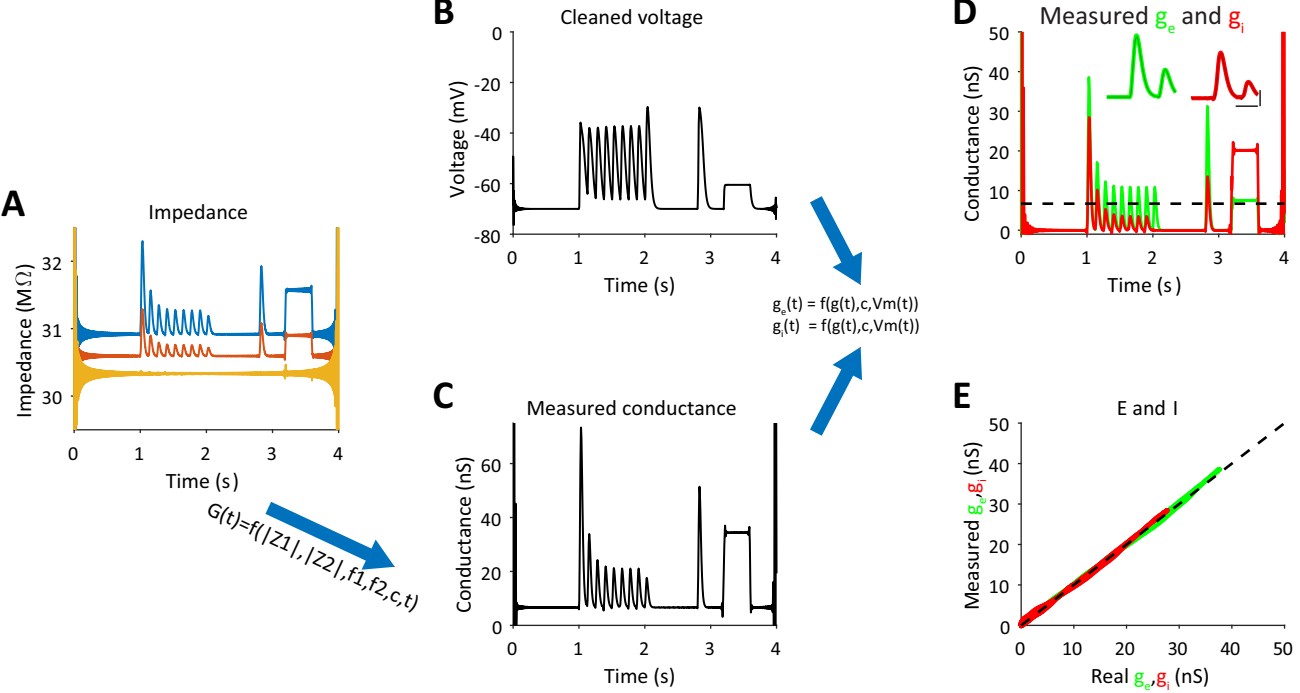

**Fig 4. Measurement of g$_e$ and g$_i$ in a single trial from impedance measurements–– simulation of a point neuron. A.** Absolute impedance curves for each frequency (as shown in Fig 2). **B.** The membrane potential after filtering out the components at the injected frequencies. **C.** Measured conductance and electrode resistance (shown in A) were obtained from the absolute impedance curves (Eqs 11, 12) and the estimated capacitance. **D.** Excitatory and inhibitory conductances were estimated from the cleaned voltage and measured conductance (Eqs 1, 2). Insets show the extended data for the first synaptic responses, superimposed with the real conductances (thin lines) (scale bars are 100ms and 10nS. **E.** Measured excitatory and inhibitory values plotted against real values that were used for the simulation.

obtain a clean version of the membrane potential. Before we use Eqs (1) and (2), we need to calculate the resting membrane potential and its corresponding leak conductance. We do this by finding the mean voltage in the cleaned membrane potential for the lower 5th percentile of the total conductance vector, which we assume reflects the resting state at which no synaptic inputs are evoked (i.e, $g_{l,est}$). The corresponding membrane potential values for this 5th percentile conductance were used to calculate the mean resting potential ($V_l$). The synaptic conductance is simply given by: $g_{s,est}(t) = g_{est}(t)−g_{l,est}$ (the difference between total conductance and leak conductance). In the transformation presented in Eqs (1) and (2), we assume that the reversal potentials of excitation and inhibition are available to us (i.e., 0mV and -70mV). The capacitance and total conductance are obtained as described above. The results of these computations are shown in Fig 4D. Our calculations revealed that the estimated conductances are almost identical to the real inputs of the simulated cell (compare Figs 3B to 4D). We note that our method allows estimating the conductances even when tonic input exits, as demonstrated in the step change in excitation and inhibition (shown between 3 to 4 seconds). In fact, the Pearson correlation between the real inputs and the estimated inputs for this simulated example were extremely high: 0.999 for excitation and 0.996 for inhibition (Fig 3E).

## Computing the excitatory and inhibitory conductances of a cell embedded in a balanced network

We asked if our approach can be used to reveal the underlying excitatory and inhibitory conductances of a model cortical neuron embedded in an active network where it receives excitatory and inhibitory inputs. Therefore, we used a simulation of a cortical network at a balanced asynchronous state [33] to obtain the excitatory and inhibitory synaptic inputs of a single cell (kindly provided by Dr. Michael Okun, University of Leicester). We used these conductances in a simulation of a single cell, in which we injected a current with two sinusoidal components (210 Hz and 315Hz) via a 50MΩ electrode and measure the response of the cell, before (Fig 5) and after filtering out the two sinusoidal components from the membrane potential (Fig 5B, black trace, which is superimposed almost perfectly with the one obtained without current injection, blue trace).

We then used our computations to estimate the excitatory and inhibitory conductances (Fig 5C and 5D). Note, however, that for both inputs the estimated conductances are more negative than expected. This is simply because the leak conductance was estimated from the 5th percentile of the total conductance of the cell, but since synaptic activity persisted throughout the trace, the leak conductance reflects a mixture of the true leak conductance and some baseline synaptic activity. Nevertheless, the estimated excitatory and inhibitory synaptic conductances were very similar to those used as inputs (Fig 5E and 5G), and similarly to the real inputs, estimated E and I conductances were highly correlated (Fig 5F). Our approach was also successful in measuring E and I inputs when they are not correlated (Fig 5H and 5J, by shifting the inhibitory input by 10 seconds relative to excitation). Indeed, as expected for this case, no correlation was measured between the measured inputs (Fig 5I). In summary, our approach allows accurate estimation of excitatory and inhibitory inputs in various conditions without any need to take into account the dynamic and statistical properties of the excitatory and inhibitory inputs.

## Measurement of E and I inputs during large variations in access resistance

Changes in access resistance due to incompletely ruptured membrane or other due to movement of the recorded cell and preparation, pressing the pipette onto the membrane, are well-known limitations of whole cell patch recordings. However, one of the advantages of the

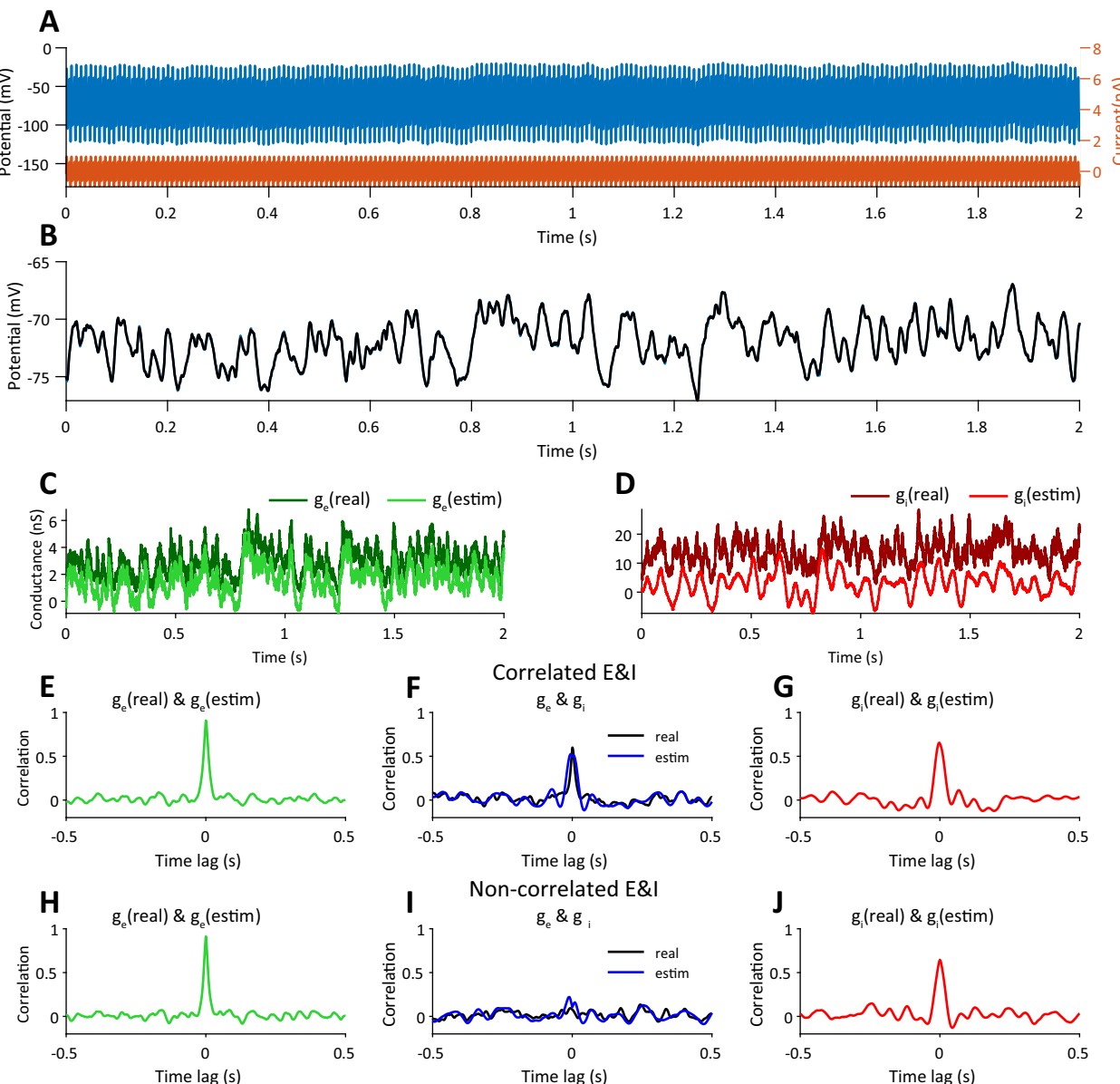

**Fig 5. Estimation of the correlation between excitatory and inhibitory conductances of a neuron embedded in balanced/unbalanced Networks. A.** Excitatory and inhibitory conductances taken from network simulations (see text) were used to simulate the response of a point neuron recorded with a 50MΩ pipette while injected with 210Hz & 315Hz 0.5 nA sinusoidal current. **B.** The voltage in A was 'cleaned' from the sinusoidal component (black trace) and it is displayed with the voltage response of the cell when no current was injected superimposed with the 'cleaned' voltage. **C-D.** Estimated excitatory and inhibitory conductances superimposed with the imposed conductances of the simulated cell. **E, G.** Color lines describe the cross-correlations between the imposed and estimated excitatory and inhibitory conductances for the balanced cortical Dynamics. **F.** The correlations between measured $g_e$ and $g_i$ (blue line) and between the imposed $g_e$ and $g_i$ (black line). **H-J.** The same analysis as for e-g but after shifting the inhibitory input by 10 seconds to mimic uncorrelated excitatory and inhibitory inputs.

approach is in its ability to track changes in the electrode and access resistance and taking them into account when calculating the total conductance of the cell with a high temporal resolution. We demonstrate it by simulating rapid changes in the electrode resistance during the in-silico recordings (Fig 6, identical synaptic inputs to those used in Fig 3). These variations led to a noisy impedance measurement (Fig 6A). However, since we can measure the access

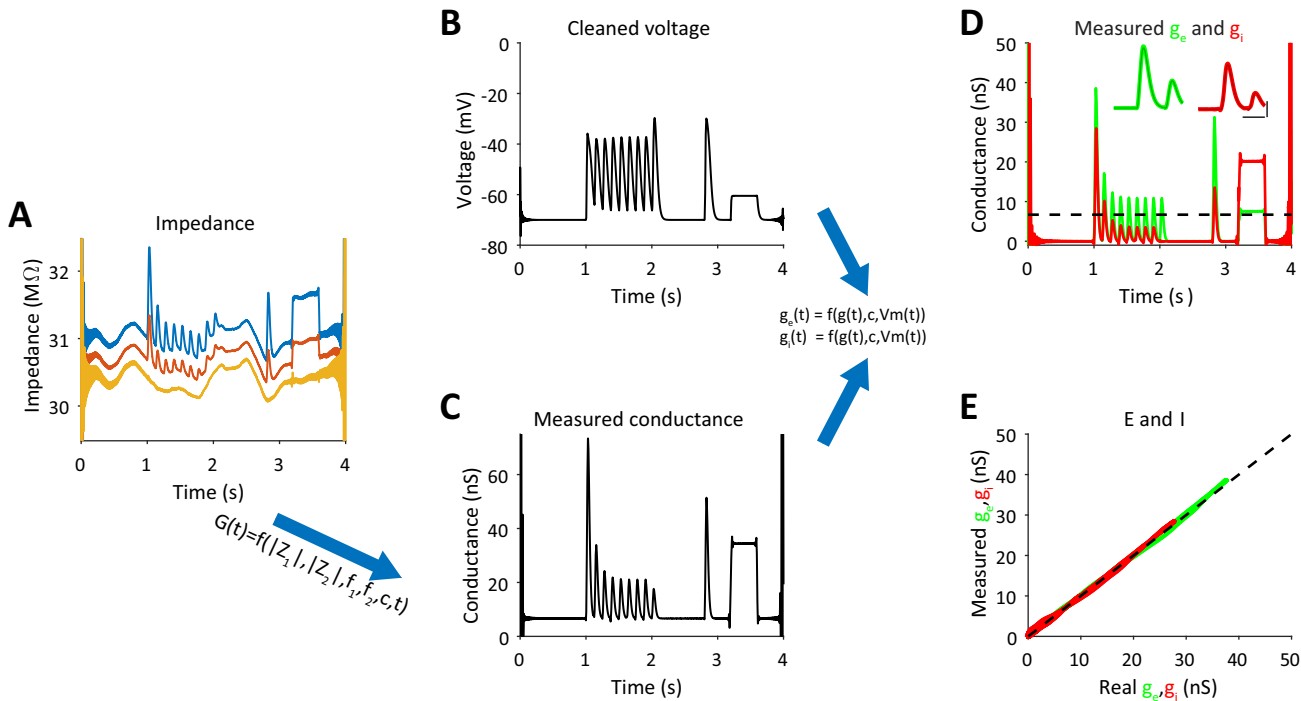

**Fig 6. Measurements of $g_e$ and $g_i$ are accurate even when electrode resistance is not stable. A.** Absolute impedance curves for each frequency depicted together with measured electrode resistance. To modulate the electrode resistance, we smoothed a lowpass Normally distributed noise signal and added it to a fixed resistance. All other parameters of the inputs are identical to those shown in Figs 3 and 4. **B-E.** Analysis of the voltage response of the point neuron as processed by the same way as shown in Figs 2 and 3. Note for the accurate estimates of $g_e$ and $g_i$ (compare to Fig 3B).

resistance over time ($R_s(t)_{est}$) and total $g(t)$ at the same time (Eqs (11) and (12)), followed by measurement of excitation and inhibition as described in Eq (2), the changes in the electrode resistance had no apparent effect on the ability to accurately estimate the inhibitory and excitatory conductances (Fig 6D and 6E).

## Measurement of E and I inputs in the presence of realistic noise

Next we asked how sensitive our measurements are in the presence of realistic noise. Therefore, we used a typical patch electrode to record a voltage trace in a slice setup when positioning the electrode outside a neuron (kindly provided by Dr. Alexander Binshtok, Hebrew University). We then added this noise to our simulated voltage prior to the measurement of excitation and inhibition (Fig 7). A sample of the voltage in the absence of sinusoidal current injection is shown in the inset of Fig 7A (voltage scale bar is 0.5 mV). Despite the presence of such noise (standard deviation of 0.04mV), and a concomitantly noisier measurement (Fig 7D) of excitation and inhibition, their values closely matched those we imposed as inputs in the simulation (Fig 7E).

## Compensation for electrode capacitance

In the above computations we assumed that the recordings are made with a pipette of zero capacitance. However, electrode capacitance can greatly affect the measurement using our novel algorithm. Most of the stray capacitance of recording pipettes is formed by the separation of the solution inside vs. outside the glass pipettes. Experimentally, it can be reduced but not eliminated by coating the pipette with hydrophobic material [34]. Pipette capacitance ($C_p$,

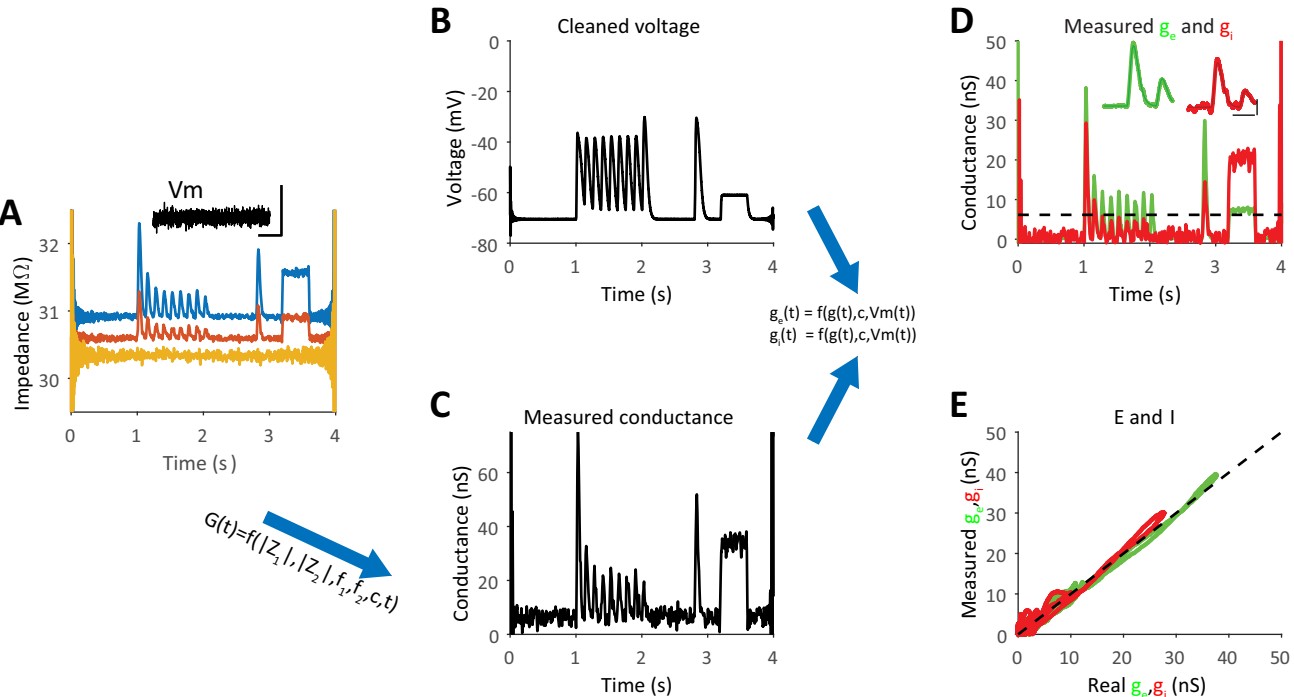

**Fig 7. Measurements of $g_e$ and $g_i$ are accurate in the presence of realistic recorded noise. A.** Absolute impedance curves for each frequency depicted together with the measured electrode resistance when real noise recorded from an in-vitro step was added to the simulated voltage before the measurement. The inset shows a voltage trace in absence of sinusoidal injection (scale bars: 0.5s, 0.5mV). All other parameters of the inputs are identical to those shown in Figs 3 and 4. **B-E.** Analysis of the voltage response of the point neuron as processed in the same way as shown in Figs 2, 3 and 6. Refer to Fig 3B for ground-truth $g_e$ and $g_i$.

illustrated in Fig 8) can also be neutralized by the electronic circuit of the intracellular amplifier, using a positive feedback circuit. In our in-silico experiment, we show that $C_p$ can greatly affect the measurement, as pipette capacitance draws some of the injected sinusoidal current. As a result, the impedance measurements for the two frequencies ($z_1$ and $z_2$) are smaller than expected from the cell and $R_s$ alone (Fig 8B, Rs is 20MΩ and the curves are well below this value). This, in turn, results in a much higher leak conductance and a completely wrong estimation in the synaptic conductances based on Eqs (11) and (12). Altogether, our estimations can be flawed, leading to negative evoked inhibitory conductance (Fig 8D).

To compensate for the impedance reduction due to the pipette capacitance we estimated Cp and then used this value to correct the measured impedances. Here we show the theoretical admittance (Y, Y = 1/Z) at each of the two frequencies for the equivalent circuit of a cell recorded with a pipette that has stray capacitance, as shown in Fig 8. The second terms in the following Equations depict the admittance of the stray capacitance (Eqs (13)–(15) were derived from the circuit that is presented in Fig 8, G is the cell's total conductance).

$$1/Z_1 = \frac{1}{R_s + \frac{1}{G+j\cdot w1\cdot C}} + j\cdot w_1\cdot C_p \tag{13}$$

$$1/Z_2 = \frac{1}{R_s + \frac{1}{G+j\cdot w2\cdot C}} + j\cdot w_2\cdot C_p \tag{14}$$

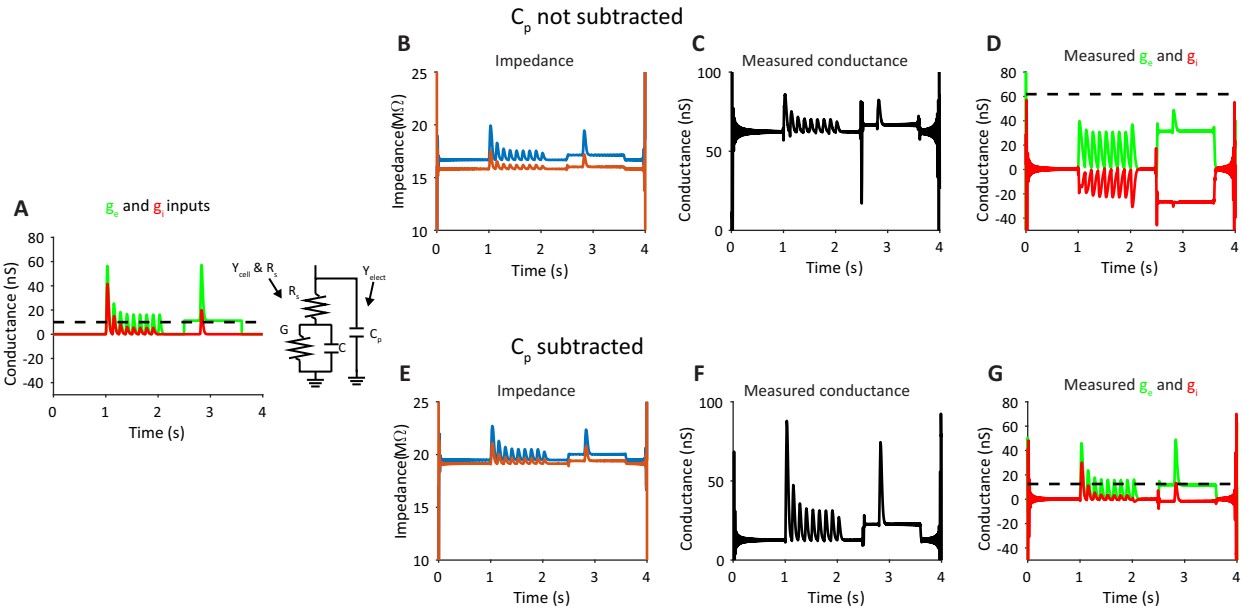

**Fig 8. Computational approach for compensation of parasitic capacitance for measurement of excitation and inhibition in an in-silico model. A.** The 'real' inputs in the in-silico experiment and the circuit representing the recording configuration. Note for $C_p$, the parasitic capacitance of the electrode ($R_s$). **B-D.** Measurement of the impedance, followed by calculation of the conductance and E and I inputs using the approach described in Figs 3 and 4. **E-G.** Similar measurements when recalculating the impedance at each frequency using Eqs (13) to (18) to subtract the effect of pipette capacitance.

From these two equations and replacing $1/(R_s + \frac{1}{G+j\cdot w1\cdot C})$ with $Y_1$ (and $Y_2$), $C_p$ is given by:

$$C_p = \frac{(1/Z_1 - 1/Z_2) - (Y_1 - Y_2)}{j(w_1 - w_2)} \tag{15}$$

However, the value of $Y$ and $Y_2$ are unknown and are those we seek. We found, however, that the second term $(Y_1-Y_2)$ can be neglected as it is much smaller when compared to the value of $1/Z_1-1/Z_2$. For example, for the parameters used in this simulation, the ratio between the latter and first terms is ~200, clearly justifying our next approximation in which we use in the measured impedance curves, as made using Eqs (9) and (10) (shown as measured $Z_1$ and $Z_2$ below, both are time dependent).

$$C_{p,est} \approx< \frac{(1/Z_1 - 1/Z_2)}{j(w_1 - w_2)} > \tag{16}$$

We then use this estimated value of $C_p$ (averaged for a selected time window (e.g., 1S) before the stimulation under the assumption that synaptic inputs are silent during this time) to calculate the estimated impedance of the cell and the electrode alone, as theoretically expected $(Z' = 1/Y' = R_s(t) + \frac{1}{g(t)+j\cdot w1\cdot C})$ which is done by subtracting from the two measured $Z$ curves the $C_{p,est}$ component following rearranging Eqs 13 and 14:

$$1/Z'_1 = 1/Z_1 - j \cdot w_1 \cdot C_{p,est} \tag{17}$$

$$1/Z'_2 = 1/Z_2 - j \cdot w_2 \cdot C_{p,est} \tag{18}$$

The new $Z'$ vectors are then used as the inputs as described above in Eqs (11) and (12) and the subsequent process as described above. This approach greatly improved the measurement of

excitation and inhibition (Fig 8E–8G). Hence, this component in the analysis, which can be switched on and off, can help resolve the analysis of real recordings, where stray capacitance always exists.

## Measuring synaptic conductances in morphologically realistic neurons

To assess how our method resolves dendritic conductances, we simulated a morphologically realistic CA1 pyramidal cell [35]. We uniformly distribute 50 inhibitory and 50 excitatory synapses proximal to the soma. We realized that due to current escape of the injected sinusoidal current to the dendrites, the estimated leak conductance is much larger than its actual value. In the case of proximal synaptic inputs, less current is escaping towards the dendrites during activation of these inputs when compared to pre-stimulation conditions. We compensated for this change by dynamically altering the strength of the leak conductance at each time point based on the estimated total synaptic conductance before calculating the excitatory and inhibitory conductances (Eqs (1) and (2)) by using this empirical equation:

$$g\prime_l(t) = g_l(1 - e^{-(g_s(t)/g_l)^2}) \tag{19}$$

Such change is equivalent to a dynamic change in the electrotonic length of cells, known to cause space clamp errors [36–38]. It shows that for weak proximal synaptic input this function strongly reduces the newly calculated leak conductance ($g\prime_l(t)$) as expected, and that this allows to compensate for the current escape. However, when the synaptic inputs get stronger the function increases the leak, as less current is expected to escape to the dendrites due to the shunting effect of the input.

Although those synapses are on average 129.92μm (±47.83μm SD) away from the soma, our method resolves the excitatory and inhibitory conductances in a single trial at least as well as the voltage clamp measurements do during two separate trials. When the synapses are moved further away, to an intermediate distance of 238.69μm (±39.71μm SD), our method underestimates the conductance to the same extent as voltage clamp (Fig 9B). Under most biological conditions synapses are not constrained to a narrow part of the dendrite. Therefore, we uniformly distributed synapses anywhere on the apical dendritic tree (Fig 9C). This resulted in synapses with an average distance to the soma of 309.92μm (±164.46μm SD). In this case, our method still follows the conductances but underperforms compared to voltage clamp. Because the measurement quality seemed to decrease with distance, we did more simulations to quantify the relationship between somatic distance and recording quality.

## Conductance measurements of proximal inputs are stable and reliable

To investigate the relationship between measurement quality and synaptic distance to soma, we simulated a single excitatory and a single inhibitory synapse at the same dendritic segment. As above, we found that we can reliably isolate the conductances when the synapse pair is close to the soma (Fig 10A). At an extremely distal synapse localization, the measurement becomes unreliable. Even the voltage clamp ceases to follow the temporal dynamics. To quantify the extent to which our measurement follows the temporal dynamics of the current we calculated the correlation coefficient between measurement and true conductance. We found that the measurements are very reliable for synapses below 400μm somatic distance (Fig 10C). Above that distance, the measurement quality breaks down abruptly for the excitatory conductance (Fig 10B and 10C).

## Discussion

We describe a novel framework to estimate the excitatory and inhibitory synaptic conductances of a neuron under current clamp in a single trial with high temporal resolution while

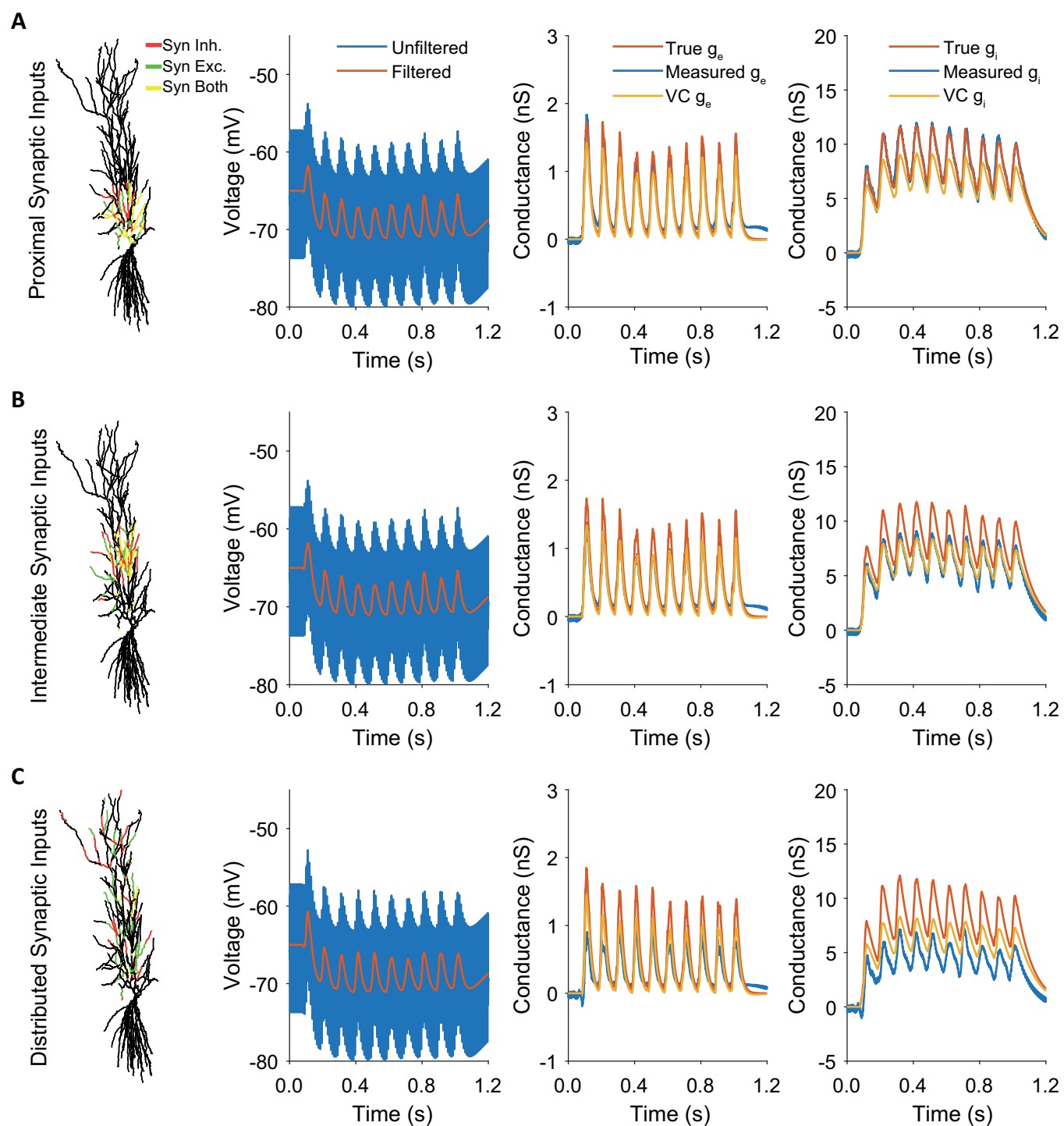

**Fig 9. Measuring conductance changes of dendritic synapses. A.** Inhibitory and excitatory synapses were placed proximal (129.92μm ±47.83μm SD) to the soma and all measurements were performed at the soma. Simultaneous conductance measurements are at least as accurate as separate voltage clamp recordings for these proximal synapses. **B.** Inhibitory and excitatory synapses were placed at intermediate distance (238.69μm ±39.71μm SD). Simultaneous conductance measurements and voltage clamp are both well correlated with the temporal dynamics but underestimate the amplitude. **C** For distributed synapses (309.92μm ±164.46μm SD) simultaneous conductance measurements underestimate the magnitude of the true conductance but still follow the time course.

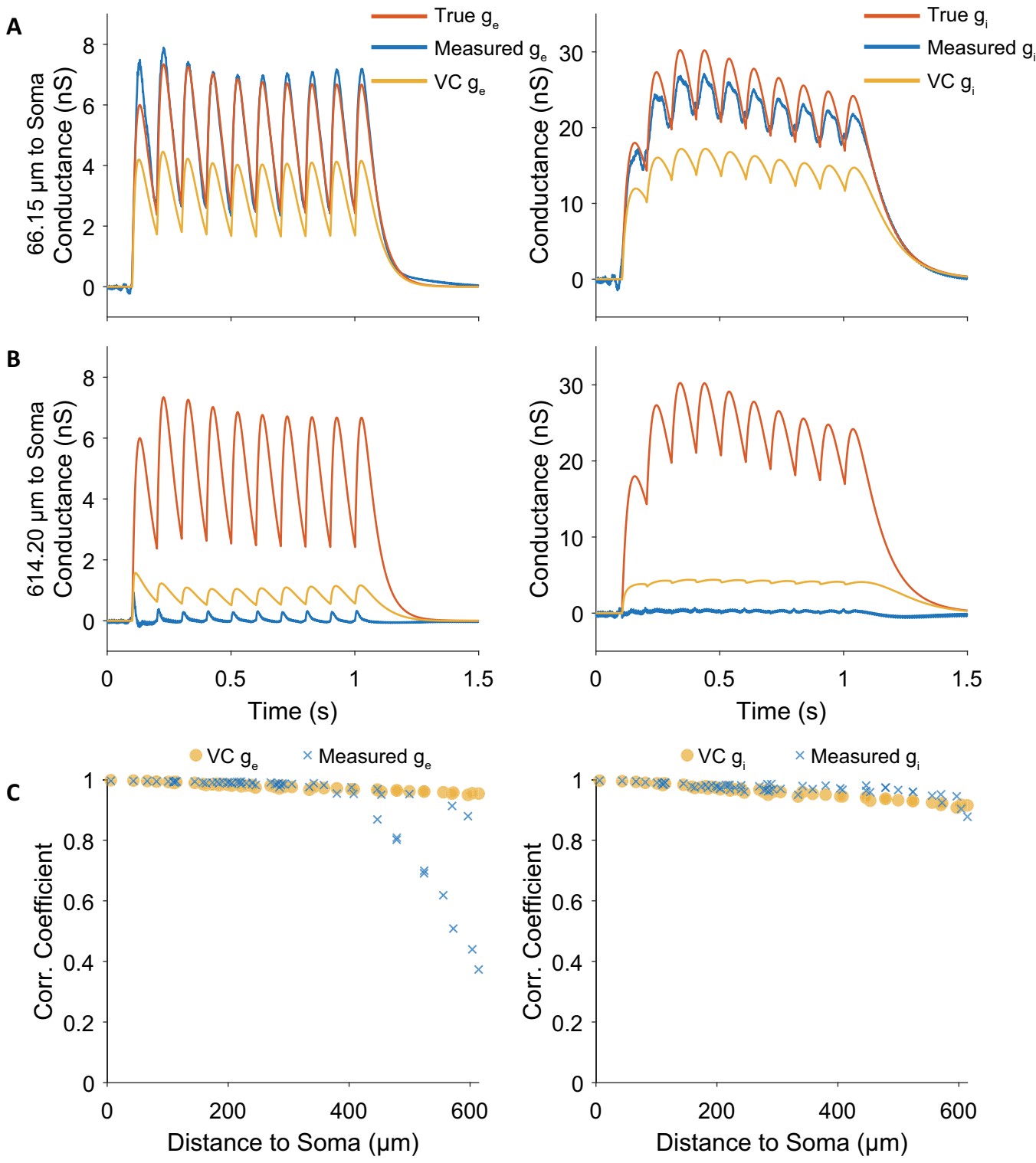

**Fig 10. Simultaneous conductance measurements are accurate for synapses up to 400μm but break down above. A.** Simultaneous conductance measurements are highly accurate for the excitatory and inhibitory proximal synapses. **B.** For extremely distal synapses, our simultaneous conductance measurements become inaccurate. Color legend as above. **C.** The correlation coefficient for many synapses confirms that simultaneous conductance measurements are at least as accurate as voltage clamp below 400μm somatic distance. For further away synapses, the conductance measurement technique breaks down abruptly for the excitatory conductance.

tracking the trajectory of the membrane potential. We show that the method allows estimating these inputs also in a morphologically realistic model of a neuron. The work described above here is theoretical and lays the foundations for future experimental work.

The method is based on the theory of electrical circuit analysis over time when a cell is injected with the sum of two sinusoidal currents. This allows us to measure excitatory and inhibitory conductances and at the same time track the membrane potential.

We demonstrated the method in simulations of a point neuron and in realistic simulations of a pyramidal cell, receiving proximal and uniformly distributed synaptic inputs. For the point neuron, we showed that we could reveal the timing and magnitude of depressing excitatory and inhibitory synaptic inputs with high temporal resolution and accuracy of above 99% (Figs 3 and 4). In another example, we used our method to reveal these inputs during an asynchronous balanced cortical state and showed that excitation and inhibition dynamics can be measured with high accuracy. Importantly, these estimations were obtained from single trials and allowed obtaining the natural dynamics of the membrane potential by filtering out the sinusoidal components of the response to the injected current. Therefore, our method is especially suitable for estimation of excitation and inhibition when these inputs are not locked to stereotypical external or internal events, such as during ongoing activity. We note that when injecting high frequency current (of a couple of hundred Hertz and above), the voltage drops mostly across the recording electrode. Here we tuned the current amplitude to produce a few millivolts sinusoidal fluctuation across the cell membrane, which should have minimal effect on voltage-dependent intrinsic and synaptic conductances when performing recordings in real neurons.

## Comparisons with other methods

*Measurement of average excitatory and inhibitory conductances of single cells*: Excitatory and inhibitory synaptic conductances of a single cell were measured both under voltage clamp or current clamp recordings, focusing *in-vivo* on the underlying mechanisms of feature selectivity in sensory response of cortical cells and on the role of inhibition in shaping the tuned sensory response of mammalian cortical neurons [6,8,39]. Conductance measurement methods were also used to reveal the underlying excitatory and inhibitory conductances during ongoing Up and Down membrane potential fluctuations, which characterize slow-wave sleep activity [40,41]. The advantages and caveats of these methods were reviewed in [29]. Common to these conductance measurement methods is the requirement to average the data over multiple repeats, triggered on a stereotypical event (such as the time of sensory stimulation or the rising phase of an Up state) and then average trials at different holding potentials. The averaged data is then fitted with the membrane potential equation (assuming that the reversal potentials are known) to reveal the conductance of excitation and inhibition at each time point. However, these methods cannot reveal inhibition and excitation simultaneously in a single trial, and only estimate averaged relationships. Our proposed method, on the other hand, allows for simultaneous measurements during a single trial. Importantly, since there is no need to depolarize or hyperpolarize the cell, our method allows measurement of synaptic conductances at the resting potential of the cell, potentially obtaining measurements of voltage dependent conductances as they progress during the voltage response to the synaptic inputs. We note that our method shares the basic approach for the analysis of point-neurons using the theory of frequency analysis of electrical circuits with capacitance measurements methods [42,43].

An alternative approach for estimating the excitatory and inhibitory conductances of a single cell was demonstrated for retinal ganglion cells [44]. In this study the clamped voltage was alternated between the reversal potential of excitation and inhibition at a rate of 50 Hz and the current was measured at the end of each step. This study revealed strong correlated noise in

the strength of both types of synaptic inputs. However, unlike the method proposed here, the underlying conductances are not revealed simultaneously and–due to the clamping–the natural dynamics of the membrane potential is entirely unavailable, preventing examining the role of intrinsic voltage dependent dynamics in the generation of neuronal subthreshold activity.

**Single trial measurements of $g_e(t)$ and $g_i(t)$ under various assumptions on synaptic dynamics.** Theoretical and experimental approaches based on the dynamics of excitatory and inhibitory conductances in a single trial were previously proposed. Accordingly, excitation and inhibition are revealed from current clamp recordings in which no current is injected. Approaches based on Bayesian methods which exploit multiple recorded trials were proposed [25] and estimation of these inputs in a single trial were also proposed but lack the ability to track fast changes in these conductances [24]. A group of other computational methods [26–28] showed that excitatory and inhibitory conductances could be revealed in a single trial when analysing the membrane potential and its distribution. Common to all these methods is the requirement to observe clear fluctuations in the membrane potential. Our method, however, allows revealing these inputs even if no change in membrane potential due to synaptic input is observed (except for the response to the injected sinusoidal current). Changes in conductance are often expected even when the membrane potential is stable, for example when a cell is receiving tonic input (see the step change in excitation and inhibition in Figs 3 and 4, between 3 to 4 seconds, resulting in a constant membrane potential value) and when a constant balance in excitatory and inhibitory currents exists.

**Paired intracellular recordings.** The substantial synchrony of the synaptic inputs among nearby cortical cells [19,21,45,46] allows continuous monitoring of both the excitatory and inhibitory activities in the local network during ongoing and evoked activities. A similar approach was also used to study the relationships between these inputs in the visual cortex of awake mice [22] as well as gamma activity in slices [4]. While paired recordings are powerful when examining the relationships between these inputs in the local network, such recordings do not provide definitive information about the inputs of a single cell. Moreover, although the instantaneous relationship between excitatory and inhibitory inputs can be revealed by this paired recording approach, the maximum inferred degree of estimated correlation between excitation and inhibition is bounded by the amount of correlation between the cells for each input, which may change across stimulation conditions or brain-state [47–49]. For example, a reduction in the correlation between excitation, as measured in one cell, and inhibition measured in the other cell can truly suggest smaller correlation between these inputs for each cell, but it can also result from a reduction of synchrony between cells, without any change in the degree of correlation between excitation and inhibition of each cell. This caveat of paired recordings prevents us from finding, for example, if cortical activity shifts between balanced and unbalanced states [50,51]. Simultaneous measurement of excitatory and inhibitory conductances of a single cell across states will allow these and other questions to be addressed.

## Limitations

Theoretically, increasing the frequency of the sinusoidal waveforms of the injected current in our method improves the temporal precision when measuring synaptic conductances. However, this comes at the expense of sensitivity, which reduces as frequency increases (Eq 3 and Fig 2). In our simulations we limited the frequency of the injected current up to about 350Hz. At this range, our simulations, depicting realistic passive cellular properties and typical sensory evoked conductance will result in a clear modulation in voltage when injecting ~1nA sinusoidal current. When bandpass filtering the voltage, the modulation is in the order of only a mV, but is still above the equipment noise.

We show that changes in access resistance due to incompletely ruptured membrane or other factors, such as mechanical vibration causing the membrane to move with respect to the pipette, can be well measured and compensated (Fig 6). Hence our approach can be implemented to estimate excitatory and inhibitory inputs of a cell in these realistic conditions.

Another aspect that might reduce the sensitivity of our method is the presence of pipette stray capacitance. We developed a modular component in the analysis that can be used to correct some of this stray capacitance (Fig 8). Importantly, no additional measurement is needed beyond the injected sine waves, done in a single trial, to measure this stray capacitance and compensate for its effect. Yet, when stray capacitance is much higher than was demonstrated here, this approach fails to provide a good estimation of the synaptic conductance. Hence, special care will still be needed to minimize any stray capacitance as much as possible.

We demonstrate in simulations of morphologically realistic neurons that we can estimate proximal synaptic inputs in a single trial using our approach. Although we underestimated these inputs when compared to simulated voltage-clamp experiments, their shape and relationships were preserved in our measurements if the inputs impinged on dendrites not more distant than 400 μm from the soma of our implementation of a pyramidal cell. Even though this limitation should be considered in real recordings, these data also suggest that the method will provide an adequate assessment of proximal inputs.

### Possible application of the method for measurement of non-synaptic intrinsic conductances

Our method can also be used when voltage-dependent conductances evolve naturally, as we can measure these inputs at the resting potential of the cell, as long as the sinusoidal fluctuations across the membrane due to the injected current are small. Such an approach therefore can be used when performing pharmacological tests, such as testing effects of modulators, agonists and antagonists of various ion channels. Due to the ability to measure these inputs in a single trial, the time course of the effects can be studied in rapid time scales while examining the effects of such drugs on both inputs at the same time.

In summary, our theoretical study shows that synaptic and other conductances can be measured at high temporal resolution in a single trial when cells are recorded at their resting potential. More research is needed to find if this approach can be used successfully during physiological recordings from real neurons.

### Feasibility of the technique in real recordings

The expected signal to noise ratio, based on the addition of realistic noise (Fig 7) is sufficiently high to measure the excitatory and inhibitory input during in-vitro recordings. However, it is clear that this framework has to be tested in real recordings of neurons. We fully disclose that we made attempts to test the method in real recordings and discovered that in most of our recordings, none shown here, measurements were unsuccessful. Following tests for impulse response of the amplifier, we found that this results from an active feedback circuit in our intracellular amplifiers. We are currently improving the amplifier circuitry and in parallel developing algorithms that will incorporate the frequency response characteristics of these amplifiers.

## Methods

### Simulations

To develop the method we constructed a simple simulation of a single compartment neuron attached to a resistor, simulating the resistance of the recording pipette ($R_s$ is the electrode

resistance). $I_m$ is the injected current and the other variables as shown in Eqs (1) and (2). Also note that the capacitive current is given by: $I_c = I_m - k \cdot (V_p - V_m)/R_s$, where $V_p$ is the recorded voltage (across the recording the pipette), $V_m$ is the voltage across the membrane only $I_c$ is stray current. For $k = 0$ we assume no stray capacitance and for $k = 1$, capacitance was included. Hence at each time point we calculated (dt is the time step of the simulation):

$$dV_m = \frac{dt}{c}\left(g_l(V_m - V_l) + g_e(V_m - V_e) + g_i(V_m - V_i) - (I_m - I_c)\right) \tag{20}$$

$$dV_p = k * dt \cdot I_c / C_p \tag{21}$$

$$V_m = V_m + dV_m \tag{22}$$

$$\text{for } k = 1, \ V_p = V_p + dV_p, \ \text{whereas for } k = 0, \ V_p = V_m + I_m \cdot R_s \tag{23}$$

To test the performance of our method in extraction of excitatory and inhibitory conductances, we simulated the response of a cell to a train of synaptic inputs which depress according to the mathematical description of short term synaptic depression (STD, [52]) with $\tau_{inact}$ = 0.003S (inactivation time constant) for excitation and $\tau_{rec}$ = 0.5S (recovery time constant) for excitation and the same inactivation time constant for inhibition (0.003S) but a longer recovery time constant ($\tau_{rec}$ = 1.3S) but exhibiting the same utilization (0.7). The values of the passive properties of the cell and the strengths of synaptic conductances in the simulation were chosen to be at a similar range of experimental data [8,14,15]. Namely, resting input resistance of 150MΩ, total capacitance of 0.15nF and pipette resistance of 30MΩ. Simulations were run using a simple Euler method with a time step of 0.1msfor all point neuron simulations except for Fig 7 (0.025ms).

## Morphologically realistic simulations

We used NEURON 7.6.7 [53] in Python 3.7.6 to simulate a CA1 pyramidal cell [35]. We loaded this cell directly into NEURON without changes to the neuron model. 50 inhibitory and 50 excitatory were distributed on parts of the apical tree. The synaptic mechanism was a modified version of the Tsodyks-Markram synapse [52] where we added a synaptic rise time (NEURON mechanism available at https://github.com/danielmk/ENCoI/tree/main/Python/mechs/tmgexp2syn.mod). The synaptic parameters are detailed in Table 1. Event frequency of both synapses was 10Hz and events were jitter with a Gaussian distribution of 10ms SD.

All measurements were performed at the soma. To simulate an access resistor in current clamp we added a section with a specified resistance between the current clamp point process and the soma. The access resistance was 10MOhm. For the stimulation current we summed two sine waves of 210Hz and 315Hz. The combined sine waves had a peak-to-peak amplitude of 1nA. Voltage clamp was performed in separate simulations with 10MOhm access resistance as during current clamp. While isolating the excitatory current, we clamped at the reversal potential of inhibitory synapses (-75mV). While isolating the inhibitory current, we clamped at the reversal potential of excitatory synapses (0mV). To convert current to conductance, we divided the current by the clamped voltage minus the synaptic reversal potential.

To investigate the relationship between measurement quality and dendritic path distance to soma, we moved a single excitatory and a single inhibitory synapse to the same dendritic section. Sections were chosen by iterating through the list of apical dendrites in steps of 5. The synaptic parameters are detailed in Table 1.

**Table 1. Synaptic parameters of morphologically realistic simulations in Figs 9 and 10.**

| Parameters for Fig 9 | | | |
|---|---|---|---|
| **Parameter** | **Value Excitatory** | **Value Inhibitory** | **Description** |
| n_syn | 50 | 50 | Number of excitatory synapses |
| gsyn | 3e-4µS | 4.5e-4µS | Synaptic weight |
| tau_rise | 1ms | 1.2ms | Rise time constant of excitatory conductance |
| tau_decay | 20ms | 100ms | Decay time constant of excitatory conductance |
| tau_facil | 0ms | 0ms | Facilitation time constant |
| tau_rec | 200ms | 600 | Recovery time constant |
| U | 0.2 | 0.4 | Utilization constant synaptic efficacy |
| Ev | 0mV | -75mV | Reversal potential |
| Parameters for Fig 10 | | | |
| **Parameter** | **Value Excitatory** | **Value Inhibitory** | **Description** |
| n_syn | 1 | 1 | Number of excitatory synapses |
| gsyn | 3e-2µS | 4.5e-2µS | Synaptic weight |
| tau_rise | 20ms | 30ms | Rise time constant of excitatory conductance |
| tau_decay | 50ms | 100ms | Decay time constant of excitatory conductance |
| tau_facil | 0ms | 0ms | Facilitation time constant |
| tau_rec | 200ms | 600ms | Recovery time constant |
| U | 0.2 | 0.4 | Utilization constant synaptic efficacy |
| Ev | 0mV | -75mV | Reversal potential |

Python simulation results were saved as.m files using SciPy [54]. Simultaneous conductance analysis and plotting were performed in MATLAB.

## Acknowledgments

I would like to thank Hagay Famini for checking and testing some computational aspects that were used in this study and to Michael Sokoletsly for outstanding comments on the manuscript. We thank Dr. Michael Okun for his comments on the early version of the manuscript and for providing the simulated data used in Fig 6 and to Dr. Alexander Binshtok and Ben Title for providing the data used in Fig 7 and their important comments on this study.

## Author Contributions

**Conceptualization:** Daniel Müller-Komorowska, Ilan Lampl.

**Data curation:** Daniel Müller-Komorowska, Heinz Beck, Ilan Lampl.

**Formal analysis:** Daniel Müller-Komorowska, Ana Parabucki, Gal Elyasaf, Ilan Lampl.

**Funding acquisition:** Heinz Beck, Ilan Lampl.

**Investigation:** Daniel Müller-Komorowska, Ana Parabucki, Gal Elyasaf, Yonatan Katz, Ilan Lampl.

**Methodology:** Daniel Müller-Komorowska, Gal Elyasaf, Yonatan Katz, Ilan Lampl.

**Project administration:** Yonatan Katz, Heinz Beck, Ilan Lampl.

**Resources:** Heinz Beck, Ilan Lampl.

**Software:** Daniel Müller-Komorowska, Ilan Lampl.

**Supervision:** Yonatan Katz, Heinz Beck, Ilan Lampl.

**Validation:** Daniel Müller-Komorowska, Ana Parabucki, Gal Elyasaf, Yonatan Katz, Heinz Beck, Ilan Lampl.

**Visualization:** Daniel Müller-Komorowska, Ilan Lampl.

**Writing – original draft:** Daniel Müller-Komorowska, Heinz Beck, Ilan Lampl.

**Writing – review & editing:** Daniel Müller-Komorowska, Ana Parabucki, Gal Elyasaf, Yonatan Katz, Heinz Beck, Ilan Lampl.

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
