## [Decision Letter · Decision Letter 0]

4 Jul 2021

Dear Ilan Lampl

Thank you very much for submitting your manuscript "A novel theoretical framework for simultaneous measurement of excitatory and inhibitory conductances" for consideration at PLOS Computational Biology.

As with all papers reviewed by the journal, your manuscript was reviewed by members of the editorial board and by several independent reviewers. In light of the reviews (below this email), we would like to invite the resubmission of a significantly-revised version that takes into account the reviewers' comments.

Although all the reviewers were generally excited about the novel method for extraction of conductances of excitation and inhibition presented in the manuscript, there were a number of major issues. Both reviewer 2 and 3 suggested that the manuscript, being a methods paper, needs testing with experimental data, and perhaps be more suited for an experimental journal rather than the computational scope of the PLOS comp biology. Based on these issue we unfortunately cannot accept your manuscript for publication in PLOS comp biology at this point. I hope you will find the comments useful.

We cannot make any decision about publication until we have seen the revised manuscript and your response to the reviewers' comments. Your revised manuscript is also likely to be sent to reviewers for further evaluation.

Sincerely,

Rune W. Berg, PhD

Guest Editor

PLOS Computational Biology

Lyle Graham

Deputy Editor

PLOS Computational Biology

Reviewer's Responses to Questions

**Comments to the Authors:**

Reviewer #1: Comments to the Authors

In this paper, authors provided a novel method to simultaneously extract excitatory and inhibitory synaptic conductances of a single neuron from the recorded membrane potential at a single trial. The method is analytical and looks very interesting. It uses two high frequency sinusoidal

Components (as injected current) and solves the general equation of subthreshold membrane potential of a single neuron for the total conductance (Gs) and the electrode resistor (Rs). From these items, and also estimating the cell’s capacitor, the proposed method infers Ge and Gi as excitatory and inhibitory synaptic conductances, respectively. To me this method is very novel. However, I vote to reject just because this method has the potential to be improved and validated with experimental data. Having said that, I should highlight that the authors provided very nice simulations to demonstrate the accuracy of the method. As well, all the points in the Discussion are appreciated. I have some major points as follow.

This method should be tested with experimental dataWhat is the impact of observation noise on the accuracy of estimated Ge and Gi?How the clean voltage is calculated? Is it filtered again?If I understood correctly, the idea of injecting two sufficiently high frequency injected currents was for estimating Gs and Rs (as well as C). Although I really liked this idea, I do not understand why authors do not inject more than two currents, for example 5 high frequency currents, and solve those conductances using a simple optimization technique (without any assumption or simplification).

Reviewer #2: # Comments to the Authors

Authors propose a frequency-domain method to infer simultaneously the excitatory and inhibitory (summated) time-varying conductances, from a single intracellular neuronal recording. As opposed to existing methods, Authors first simulate a single-trial current-clamp stimulation in a a passive RC circuit, while recording its electrical potential. They later explore the performance of the method in another numerical simulations, employing a multicompartment biophysically realistic CA1 model cell.

I disclose that I must have seen an earlier and more primitive exposition of this idea as a 2019 bioarXiv entry. It greatly captured my interest as an experimentalist. I think that the concepts behind the method are definitely of immediate interest for the (more) specialised audience of a methods journal (e.g. Neuron, J. Neurosci. Method) - even though the method has not (yet) been demonstrated by the Authors in an experiment - and less for the audience of a Comp Neurosci journal.

I recommend major revisions.

## Major points

1)

The presentation of the manuscript could definitely be improved. Math notation and precision in using throughout the text is not "stable" and should be revised carefully.

Especially the description of the math could be made a bit clearer, taking a bit more of presentation, discussion and analysis. Some of the steps skipped (definitely accessible for a reader with a math or engineering background) might make a naive reader lost.

2)

I recommend an improved exposition of the hypotheses behind the use of frequency-domain method, with quasi-stationary parameters. Strictly speaking, eq. 3 is obtained from eq. 1, upon Fourier transformation, but a sort of "separation of time-scales" has been invoked, adopted, but not illustrated. While Fig. 2 is obtained keeping g_e and g_i fixed in time, eq. 3 assumes they can vary. If they do vary with time, then Fourier-transforming eq. 1 is no longer straightforward in the frequency domain.

I see of course that this does work in simulations, as a reverse method, but it still lacks of a justification and a more extensive discussion.

3)

The choice of the sample values of the electrode resistance (30 and 50 MOhm) seems unusual at a first sight. Could it be that it includes the access resistance? This is mentioned nowhere and it seems anyway not typical in "good" (in vitro) recordings. Perhaps do Authors refer to access resistance in vivo? Please explain and give some references to the literature.

Does your method for estimating the capacitance have similarities with this classic 1988 paper by Neher (10.1007/BF00582306) and this 1997 extension (10.1016/S0006-3495(97)78810-6), using two sine waves ? A discussion of similarities and differences with past approaches might be useful for the curios reader.

Could the approach of Badel et al. (10.1007/s00422-008-0259-4) be of any relevance for the capacitance estimation (see their Fig. 1c)?

4)

Can you comment on the opportunities/obstables represented by online parametric (concentional bridge balance and capacitance neutralisation circuitry) and (offline) non-parametric (as in the Active Electrode Compensation of Brette and colleagues - see 10.1016/j.neuron.2008.06.021)?

Particularly the latter method should (theoretically) capture the impulse response of pipette and amplifier. That might ameliorate your implementation problems.

5)

I would make it more explcit that you are no longer considering any off-sets in the membrane potential and that all the quantities are referred to the "effective" resting membrane potential.

However this "effective" resting potential is, strictly speaking, not constant as it is equal to:

(gl Vl + gE(t) Ve + gI(t) Vi) / (gl + gE(t) + gI(t))

Can you clarify if and why these effects are negligible in your hands?

6)

An expansion of the discussion on "Feasibility of the technique in real recordings" is strongly suggested.

As the Authors are aware, this is the strongest selling point of the method.

Some aspects related to (old) patch-clamp amplifier designs criticism (see 10.1016/S0006-3495(98)74007-X and 10.1016/s0166-2236(96)40004-2) might be perhaps of use.

## Minor points:

1)

I suggest using (in general) "j" for the imaginary unit, to distinguish it from the current.

I also suggest expressing eq. 3 as

Z(w) = R_s + 1/(G_s + j w C)

rather than rearranging terms to get real_part + j * imaginary_part. It will resonate immediately with people familiar with RC filters. Calculation of magnitude and phase can be anyway done quickly by the properties of the "ratios and products of complex numbers".

2)

Throughout your manuscript, you considered a rather strong current injection (~1-2 nA large). I would make it clear that these are not DC stimuli, leading certainly to spiking activity. They are AC stimuli, like those employed for the analysis of subthreshold membrane resonance (see the classic work by Nelken and colleagues).

It is important to stress that the frequency is so high (compared to the membrane subthreshold cut-off frequency) that the membrane potential will be extremely attenuated and thus (almost surely) not recruiting any voltage-gated intrinsic current.

3)

The math notation is (very) often non-uniform throughout the text, or a bit imprecise.

- page 4, eq. 1 and line 8: please use (V_l, V_e, V_i) or (V^l, V^e, V^i), not both.

- page 4, eq. 1, please make it clear that g_e and g_i (and g_s) are, together with V, all functions of time: make the "(t)" dependency explicit.

- page 4, eq. 1: although conventional, the current "I" should have a positive sign on the right hand side of the equation, assuming as positive the current "entering" the cell;

- use, throughout the manuscript and figures, "f_1" and "f_2" (subscripted indexes) for the frequencies instead of "f1 and f2"sometimes and "F1 and F2" some other times.

- use, throughout the manuscript and figures, "g_e" and "g_i" (subscript), instead of sometimes using "G_e" and "G_i".

- page 7 (text and eq. 4): please use subscript for pulsations "w1", "w2", and frequencies "f1","f2", as you do it for the two components amplitude (i.e. I_1 and I_2).

4)

Some text and figures contain typos.

- page 4, section title: check the extra "a";

- page 8, check the spelling of "experimentalis";

- all figures: please increase your care for the "spacing" and "layout" of the subpanel titles: they are sometimes wrongly displaced and can get confusing.

- Fig. 1b: check spelling of "Membrane potentia"

- Fig. 2, axes: please use the same symbol for frequency "f" instead of "Fr" and "Z" instead of "imped";

- Fig. 2, use a single convention for indicating the units ("..., MOhm" versus "Phase(rad)")

- Fig. 2a,d: for consistency, use abs(Z) and phase(Z).

- Fig. 2c: the black line for "data1" has not been included

- Fig. 2c: the circuit sketch is "partly occluded" by the legend;

- Fig. 2b,d: use "g_s" instead of "cell G", for consistency with the text;

- Fig. 4c (and 6c): check spelling of "Measured Conductane";

- Fig. 4d,e (and 6d,e): use the same notation "g_e" and "g_i" and not another one as "E" and "I".

- Fig. 4a,d (and 6a,d): check the x-axis label (removing extra space before the closed parenthesis)

Note that the caption of Fig. 3 neither refers nor illustrates panels e.

Note that the caption of Fig. 5 neither refers nor illustrates panels g,h,i,j.

Note that the caption of Fig. 6 contains a typo (i.e. "lowpassong").

5)

- Fig. 2b and d are difficult to grasp quickly;

- Fig. 3f,g,h,i contain almost no information, at the choosen level of representation; consider replacing them by a block diagram, with what is input and what is the filtering/analysis operation.

6) eq. 6: can you use "arctan()" instead of tan^-1 ? It would improve the clarity.

7) Can you explain why you had to use Hilbert and could not use FFT ?

Reviewer #3: Neurons’ activity is by far determined by the interplay between the activity excitatory and inhibitory synapses on their membrane. The effect of an active synapse is to temporarily change the membrane conductance and selectively permit the current of specific ions in or out of the cell. It is thus essential to measure synaptic inputs in neurons to understand how they transform their input into an output. However, previous methods could only allow the measurement of either the excitatory or inhibitory conductance at a time. These methods were helpful in studying the response to a repeating sensory input such that the measurement could be switched from measuring excitation to inhibition. But this could not reveal their co-variation. Neither could it resolve their co-activity when there was no specific stimulus to trigger upon.

This manuscript introduces a novel protocol and analysis method to resolve the excitatory and inhibitory conductances impinging on a neuron simultaneously in a single trial. The study cleverly harnesses the resistance of the measuring electrode (which is often considered a problem/bug). It shows that the combined response of the neuron-electrode system to sinusoidal current injections can be used to resolve the excitatory and inhibitory conductances simultaneously. This is a significant improvement on any existing method and was actually considered an unsolvable problem until now.

The manuscript is very methodological, starting with the simplest case and gradually adds a various level of complexity, solving most of the relevant cases, including variations in the electrode access resistance, electrode capacitance, and synaptic inputs located on a dendritic tree.

I strongly support the publication of the manuscript. I have no major comments on the manuscript. All of my comments relate to clarification of the text. I feel that the paper should take a more careful approach towards the notations. Some of the variables mentioned appear in different equations with different meanings, and this is difficult to keep track of.

General comments:

1. The text on page 8 is unclear. It goes between general statements and specific examples and, in general, is not very methodic. Also, the derivation of equation 8 is not clear at all.

2. In addition, the issue of capacitance compensation is not mentioned at this stage of the text (appears later), while clearly, this is a significant factor as the capacitance of the pipette is much larger than that of the cell. It would be wise to mention here that there is a challenge, and it will be treated later (such that the reader does not think it is ignored).

3. If I understand correctly, equation 5 basically represents the fact that the decay of the impedance is linear with w*C on the log-log plot, as seen in the plot of figure 2a. However, when the electrode is present, this linearity breaks down, as the authors demonstrate on 2c. So is the approximation in eq.5 really justified?

4. The choice of the two fundamental frequencies 210Hz and 315Hz seems somewhat arbitrary. Clearly, if they are too similar there is a separation problem and if one is too high or the other too low there will be interference with the capacitance and resistance. Is there some optimization process that can be suggested to pick the “best” frequencies based on the cell capacitance and the electrode resistance?

5. Equations 11 & 12:

a. The notations of Equations 11 and 12 are not fully clear. The equations use:

i. “c” - which is the estimated capacitance (but the estimation is not used in the symbol)

ii. “Rs” – not clear which Rs is that, the real one, the estimated one from the first procedure (i.e., eq. 6?)

b. So, for example, which is the Rs that appears in equation 12 – is it Rs from equation 11 or is it Rs from equation 6.

c. In addition, in the text right after the equation it refers to Rs_{est2} and G_{est2} in equations 11 and 12 which do not really appear there. These notations should be revised.

d. I might be wrong but if you use delta_w for (w1-w2) it seems that eq. 11 could be simplified.

e. Do the authors have an explanation why G_{est} is only dependent on z1 and w1 and not on z2 and w2? If so, it would be nice to have it in the text following eq. 12.

6. Capacitance compensation: Again, notations are not confusing. The authors define Y=1/Z and then in equation 13 they define Y1 as a different quantity which is no 1/Z so when in eq. 15 they use 1/z1 and Y1 as two different quantities it turns out to be very confusing. This is amplified in the discussion around eq. 17 and 18 where again there is a use in Rs which is used in the definition of Y but also in eq. 11 & 12 (see above).

7. Dendritic input – eq. 19 comes as a surprise with no real justification. It also feels somewhat circular because the estimation of g_s depends on g_l. It would be nice if the authors explain how did they come up with this equation and how do they actually use it.

8. Dendritic inputs – It is well known that the conductance of distal dendritic input cannot be resolved precisely due to space clamp issues. It would be interesting to discuss here the “effective” conductance of the distal inputs. To some extent while the algorithm might not be able to restore the original conductance in the dendrite it might give a good estimate of what this conductance looks like from the soma.

9. In that context consider adding citations to:

a. Häusser M, Roth A. Estimating the time course of the excitatory synaptic conductance in neocortical pyramidal cells using a novel voltage jump method. Journal of Neuroscience. 1997 Oct 15;17(20):7606-25.

b. Rall W, Segev I. Space-clamp problems when voltage clamping branched neurons with intracellular microelectrodes. In Voltage and patch clamping with microelectrodes 1985 (pp. 191-215). Springer, New York, NY.

Minor comments:

The writing could be improved. For example on pp. 3 “Hence, as these methods provide only an average picture and thus fail to capture the instantaneous and trial-by-trial based insight in the relations between excitation and inhibition.”

Figure 1 – there is no legend on the figure showing which color is which.

The pass from page 5 to 6 is not obvious. It would be better to show how you are going to use the impedance in theory and only then add the pipette resistance. Right now the text seems to answer a question that wasn’t even asked at that point.

Legend of figure 2b is sloppy.

Pp6,7 – it would be good to show on a separate panel the intersection point as a function of impedance to support the claim.

P8: Typo: “experimentalis”

P10: “(Figs. 3f to i, due to screen resolution are as patches of colors)” – looks like there is a word missing: “are displayed as patches…”?

Legend of figure 5 could be improved.

Fig 8. It would be better to reverse the order of the labels (i.e., True g_e, measured g_e, VC_g_e)

**Have the authors made all data and (if applicable) computational code underlying the findings in their manuscript fully available?**

Reviewer #1: Yes

Reviewer #2: Yes

Reviewer #3: Yes

PLOS authors have the option to publish the peer review history of their article (what does this mean?). If published, this will include your full peer review and any attached files.

Reviewer #1: No

Reviewer #2: No

Reviewer #3: No
---

## [Decision Letter · Decision Letter 1]

5 Nov 2021

Dear Prof. Lampl,

Thank you very much for submitting your manuscript "A novel theoretical framework for simultaneous measurement of excitatory and inhibitory conductances" for consideration at PLOS Computational Biology. As with all papers reviewed by the journal, your manuscript was reviewed by members of the editorial board and by several independent reviewers. The reviewers appreciated the attention to an important topic. Based on the reviews, we are likely to accept this manuscript for publication, providing that you modify the manuscript according to the review recommendations.

Regarding the reviews, we believe that the suggestion to include a "case report" is an excellent idea, and would strengthen the impact of the work. Nevertheless, we leave it up to you whether to include that in the present manuscript, and thus submit a new revision, or instead consider a second publication in a different journal to that effect. If you choose not to make the changes, that is fine and you can submit the paper and we will process it as is. 

Sincerely,

Lyle J. Graham

Deputy Editor

PLOS Computational Biology

Lyle Graham

Deputy Editor

PLOS Computational Biology

[LINK]

Reviewer's Responses to Questions

**Comments to the Authors:**

Reviewer #1: Thanks for your responses. All of my previous major and minor points - except that related to the experimental validation - were addressed. As well, the authors improved the quality of the paper significantly.

I can agree with the statement that " There are

numerous examples from physics, in which theories for measurements were established long

before technologies matured enough to allow full implementation of the method".

However, I want to share this idea with the authors that some detailed discussions related to your recent experience with experimental data (through your new collaboration) can be very useful for the community. It would be great if authors add a "Case Study" in the manuscript and discuss your recent results. Capitalizing on these points in Discussion (I believe you made most of them in your current manuscript) can be significantly helpful for other researchers using/trying your method.

As I am not aware of practical things with respect to this collaboration and the extent to which you can share some data and figures, I leave this idea to authors to check its possibility.

Reviewer #2: Authors have satisfactorily addressed my concerns. I thank them for their effort and I hope we might even be able to collaborate on the actual experimental validation of their interesting novel method.

Reviewer #3: The authors addressed all my comments. I am satisfied with these changes and I think that this revision is much improved. I recommend publishing.

**Have the authors made all data and (if applicable) computational code underlying the findings in their manuscript fully available?**

Reviewer #1: Yes

Reviewer #2: Yes

Reviewer #3: Yes

PLOS authors have the option to publish the peer review history of their article (what does this mean?). If published, this will include your full peer review and any attached files.

Reviewer #1: **Yes: **Milad Lankarany

Reviewer #2: **Yes: **Michele GIUGLIANO

Reviewer #3: No

Figure Files:

Data Requirements:

Reproducibility:

References:

---

## [Editor Report · Decision Letter 2]

6 Dec 2021

Dear Prof. Lampl,

We are pleased to inform you that your manuscript 'A novel theoretical framework for simultaneous measurement of excitatory and inhibitory conductances' has been provisionally accepted for publication in PLOS Computational Biology.

Best regards,

Rune W. Berg, PhD

Guest Editor

PLOS Computational Biology

Lyle Graham

Deputy Editor

PLOS Computational Biology

---

## [Editor Report · Acceptance letter]

21 Dec 2021

PCOMPBIOL-D-21-00712R2 

A novel theoretical framework for simultaneous measurement of excitatory and inhibitory conductances

Dear Dr Lampl,

I am pleased to inform you that your manuscript has been formally accepted for publication in PLOS Computational Biology. Your manuscript is now with our production department and you will be notified of the publication date in due course.

With kind regards,

Agnes Pap
